# Physical Activity, Sedentary Behaviours and Duration of Sleep as Factors Affecting the Well-Being of Young People against the Background of Environmental Moderators

**DOI:** 10.3390/ijerph16060915

**Published:** 2019-03-14

**Authors:** Dorota Kleszczewska, Agnieszka Małkowska Szkutnik, Jadwiga Siedlecka, Joanna Mazur

**Affiliations:** 1Institute of Mother and Child Foundation, Kasprzaka No. 17a, 01-211 Warsaw, Poland; 2Department of Biomedical Foundations of Development and Sexology, Faculty of Education, Warsaw University, Mokotowska No. 16/20, 00-561 Warsaw, Poland; amalkowska@wp.pl; 3Department of Work Physiology and Ergonomics, Nofer Institute of Occupational Medicine, Św. Teresy od Dzieciątka Jezus No. 8, 91-348 Łódź, Poland; jadzias@imp.lodz.pl; 4Department of Child and Adolescent Health, Institute of Mother and Child, Kasprzaka No. 17a, 01-211 Warsaw, Poland; joanna.mazur@imid.med.pl

**Keywords:** well-being, adolescents, physical activity, sleep, sedentary activities

## Abstract

Mental health problems during adolescence are becoming increasingly frequent. The purpose of this study was to evaluate the total impact of selected behavioural and environmental factors on the variability of mental well-being indexes of young people aged 15 to 17 years. The survey, conducted as part of the last round of the Health Behaviour in School-Aged Children (HBSC) 2017/18 study, covered 3693 secondary school students in Poland at the average age of 16.53 years (SD = 1.09). Dependent variables: depression (CES-DC), stress (Cohen scale), satisfaction with life (Cantril’s Ladder), and self-efficacy in social relations (Smith and Betz scale). Independent variables included: physical activity; sedentary behaviours; length of sleep; and perception of the environment in which the respondent was raised. The analyses were adjusted by gender, age, and occurrence of chronic disease. It was demonstrated that gender, duration of sleep, and perception of the surrounding environment proved to be significant predictors of all four mental health indicators. The protective influence of physical activity appeared to be a particular advantage in a less-supportive environment. The intervention programmes aimed at improving the mental well-being of young people should include promoting physical activity and sufficient sleep; furthermore, environmental moderators should be taken into consideration.

## 1. Introduction

Mental health problems affect 10–20% of children and adolescents. The results of meta-analyses indicate a high discrepancy between measures taken to promote mental health and prevent problems among adolescents, on the one hand, and this group’s actual needs, on the other [1]. The connection between mental health disorders and other areas of health and functioning, such as school attainment, addictions, and sexual risk behaviour, is becoming evident. It is important to define protective factors as well as risk factors for the mental health of the developing population and, further, to take biological, psychological, and social determinants into account when so doing. Among the social determinants, apart from the family and school, an important factor is the local community, the disintegration of which may increase the risk of mental health disorders in this vulnerable population [2]. 

Physical activity (PA) is a generally recognised protective factor [3]. Unfortunately, despite coordinated activities of various agencies responsible for public health, the level of PA in society has not been improving; in fact, it is even periodically deteriorating [4]. For intervention programmes to be effective, it is necessary to understand the differences resulting from various age-, gender- and environmental-based factors [5]. 

Insufficient PA is particularly dangerous in the second decade of life. Its absence has a negative effect on the general physical development and health of young people; in addition, it exacerbates problems of a mental nature, as often emerge during adolescence. According to the systematic survey of Bor et al. based on 19 publications, it was found that the scale of deterioration of the mental health of teenagers began to rise in the 21st century, especially in relation to internalisation disorders among girls [6]. 

Despite the generally recognised positive role of PA, the percentage of young people who adhere to World Health Organisation (WHO) guidelines for exercise remains at a persistently low level. Three results of the Health Behaviour in School-Aged Children (HBSC) studies from the 2013/2014 school year indicate that the recommendations regarding moderate PA (a minimum of 60 min a day) in a combined international sample of 42 countries (220,000 students aged 11 to 15) were fulfilled only by an average of 20.4% of respondents. The results of the Polish HBSC studies indicate a decline in the percentage in this age group from 24.9% in 2014 to 17.2% in 2018 [7]. At this point, it is difficult to assess whether this declining trend will also be observed in other countries or whether Poland’s position in international rankings will deteriorate significantly (from the good 11th place in 2014).

More and more frequently, authors writing about the PA of teenagers take into account various forms of expending energy, without limiting them to traditional forms of intensive or moderate PA and/or participation in sports. This includes active transport to school, helping with household chores, and various forms of entertainment involving PA—including even Kinect-type console games that simulate sports [8,9]. In addition, research concerning the PA of adolescents should include information about all aspects of the absence of such activity, including sedentary activities (e.g., watching television or using a computer or mobile devices). Another important issue is the duration of sleep; indeed, the impact of this factor for proper physical, mental, and social development is being increasingly recognised [10]. Children aged 6 to 12 years should sleep a minimum of 9 h a day, and teenagers aged 15 to 18 years should obtain at least 7.5 h of sleep. An adequate amount of sleep ensures the appropriate level of hormones responsible for growth and reduces the risk of obesity, high blood pressure, and insulin resistance among young people [11]. Based on American studies, only 9% of adolescents fulfil these recommendations [12]. Moreover, according to a literature review conducted by Sounders et al. examining 13 publications based on an overall group of over 36 thousand children aged 5 to 17 years, PA, lack of sleep, and sedentary activities interact with each other to produce even greater negative effects.

The above-mentioned three behavioural factors may also be affected by ecological determinants [13], economic deprivation and social capital in the neighbourhood, the socio-economic status of the family, and the level of support given to the child by key persons in his or her life [14]. First, poverty in the family is an additional source of stress and is connected with limited access to sports in free time. Second, the authors of many studies have found that sharing a bed or bedroom with siblings or parents has a significant adverse effect on the duration and quality of sleep among young people [15]. On the other hand, a TV set, computer and audio-video equipment in a teenager’s room also constitutes a sleep disrupting factor [16]. 

Despite such compelling findings, there seem to be no studies which consider these three factors—PA, sedentary activities, and duration of sleep—in relation to the spectrum of indicators of mental well-being among young people. In addition, there are no studies that examine the above correlations in the context of an objective analysis of the state of health and environmental determinants, as they may play the role of moderators.

## 2. The Purpose of the Study

The purpose of the study was to evaluate the impact of select behavioural factors and features of the surrounding environment on the variability of specific mental well-being indexes among adolescents aged 15 to 17 years. 

The following research questions were formulated:
Do demographic features, the objectively evaluated state of health, and general characteristics of the surrounding environment have an influence on the mental well-being of young people?To what extent do various forms of PA, sedentary behaviours, and the duration of sleep affect the mental well-being of young people?Do demographic features, objectively evaluated state of health, duration of sleep, and general characteristics of the surrounding environment moderate the relationship between PA or its deficits and the mental well-being of young people?


## 3. Materials and Methods

### 3.1. The Studied Persons

The survey, conducted during the 2017/2018 school year, incorporated 3693 Polish students who were asked to complete the extended HBSC questionnaire. The group included 47.1% girls and 52.9% boys. Of the surveyed sample, 60.2% lived in cities, and 39.8% resided in the countryside. Participants were aged 14 to 18.5, the average age being 16.53 years (SD = 1.09). Of the studied group, 54% were third-grade junior high school students (118 school classes from 94 schools) and 46% were second-year secondary school students (84 classes from 55 schools). The older age group included only general education secondary schools (60% schools) and technical secondary schools (40%); other types of secondary schools (e.g., vocational) were excluded. More detailed information about the organisation of this round of HBSC studies can be found in the national report. The survey questionnaire and the research organization scheme were approved by the Bioethical Commission operating at the Institute of Mother and Child (opinion number 17/2017, 30 March 2017). Students had the right to refuse to participate in the study or not to answer questions that seemed too sensitive to them [17]. 

### 3.2. Applied Tools

Four scales included in the HBSC 2017/2018 Study Report, as proposed by the Health and Well-Being Focus Group for that report, were adopted as dependent variables. These were selected from among the scales describing the negative and positive aspects of mental health, three of which were optional packages chosen by the Polish team. 

Depressive symptoms and level of stress among teenagers were analysed as negative aspects of mental health. A high level of both indexes was considered to indicate worse results.
**Depression** was measured using the Depression Scale for Children (CES-DC). This is a well-established and proven instrument for defining mood disorders in both adults (in that respective form) and teenagers. The original Centre for Epidemiologic Studies Depression Scale (CES-D), designed for adults, consists of 20 items addressing various depressive symptoms occurring during the previous week; most statements focus on the affective component of the disorder [18]. An abbreviated version, designed for children and limited to 10 items, was attached to the HBSC Report. All items inquire about the frequency of symptoms, which are rated on a 4-point Likert scale. Four categories of answers are provided, from *rarely or never* to *all the time*, which corresponds to a frequency of less than 1 day a week to 5 to 7 days, respectively. Eight of the items have a negative, and two have a positive, scoring; for the purposes of this survey, the latter two items were re-coded [19]. The scale cumulative score ranges from 0 to 30 points. A higher score reflects a greater degree of depression symptoms. For the collected data, the CES-DC scale has a bivariate structure and is consistent, with a Cronbach’s alpha of 0.846.**Stress** was measured using Cohen’s Perceived Stress Scale (PSS). This instrument has four items: two positive and two negative. Answers are provided along a Likert-type scale, from *never* to *very often*; the range of the scale is from 0 to 16 points, higher scores indicate more or less distress. The items, among others, concern a feeling of nervousness or irritation and loss of control over one’s life [20]. In the analysed sample, the consistency of the PSS was assessed at a Cronbach’s alpha level of 0.745; its univariate structure was demonstrated.The positive aspects of mental health among young people were measured using the indexes of general satisfaction with life and self-efficacy in social relations. Higher values indicate better results.**General satisfaction with life** was defined on the basis of Cantril’s Ladder, which was confirmed as reliable by HBSC studies [21]. Cantril’s Ladder, or Scale, is a simple, one-element visual scale (range 0–10). It is the only measure in the set of adopted dependent variables which is included in mandatory questions in the HBSC report.**Self-efficacy**. The Scale of Perceived Social Self-Efficacy (PSSE) was developed by Smith and Betz [22]. It is designed to measure the level of self-confidence in various social situations. The eight positively-oriented questions start with the words ‘How well can you’. Five categories of answers were used in the version adapted for the HBSC Study Report; only extreme possible responses had the labels *not at all* and *very well*. The range of the scale is 0–32 points. The analysed sample confirmed its univariate structure and high internal consistency (Cronbach’s alpha = 0.841).When defining the **independent variables**, the researchers decided to reduce the information to be analysed and develop four combined indexes addressing PA, sedentary activities, duration of sleep, and features of the surrounding environment. In three cases (excluding sleep), standardised indexes were developed using Principal Components Analysis (PCA). By definition, the standardised indexes had an average value of 0 and standard deviation of 1, while continuity of the scale enabled a precise division of the population. A division of 20:60:20 was adopted. The categorised indexes were labelled low, average, and high. With respect to the surrounding environment, the extreme ranges were defined as unsupportive and supportive. With respect to the distribution of figures in the population, it must be remembered that—apart from sleep duration—it was an arbitrarily adopted relative distribution.The overall PA index was based on the Moderate to Vigorous PA (MVPA) indicator, derived from the Prochaska screening test [23]; the VPA indicator (intensive PA) concerning leisure time activity [24], and two questions concerning participation in organised sports activities (of both team and individual nature). The MVPA and VPA measures were re-coded into four categories, and participation in sports activities was coded dichotomously (yes/no). The combination of these questions into one indicator is supported by the univariate structure of the scale, although the level of consistency arouses some reservations (Cronbach’s alpha = 0.601).The overall sedentary activities index (used in HBSC protocol since 1985/1986) took into account the time devoted daily to watching films, playing computer games, and other computer activities (e.g., mobile devices). It was calculated as a weighted average of school days and weekends. The univariate structure of the scale supports combining the questions into one indicator, but the consistency level was too low (Cronbach’s alpha = 0.576).The overall assessment index of the surrounding environment was developed using a two-phase method; it consisted of a total of 18 questions. As a first step, four scales were derived: neighbourhood deprivation, neighbourhood social capital, social capital of the school, and parental support. An additional index, measured along a visual scale, referred to the social position of the family. A description of these partial scales is found in Table 1. As a second step, the measures were combined into one index—which is supported by the univariate structure of the scale (although, here also, the consistency level proved to be too low; Cronbach’s alpha = 0.526).Duration of sleep was calculated on the basis of the declared time of going to bed and getting up on school days. Three categories were differentiated: up to 6 h, 6.5–7.5 h, and 8 h or more. Among these groups, the distribution of the respondents was 23.5%; 49.5%, and 27.0%, respectively.Controlled variables were as follows; gender; age (continuous variable), used interchangeably with school grade (two categories); and occurrence of a chronic disease, reported by the student, or other long-lasting health problems confirmed by a physician (17.6% of respondents). 


### 3.3. Methods of Analysis

According to the Shapiro–Wilk test, the distribution of the four scales assessing mental well-being and the four scales relating to behavioural and environmental determinants were highly significantly different from normal (*p* < 0.001). Four variables were analysed in parallel without mutual connections. In the univariate statistical analysis, nonparametric methods were used, and in the multidimensional analysis, a generalised linear model was applied. The third-degree interaction: gender * general surrounding environment index * general PA index was considered the most important and consistent with the topic of the study. A number of auxiliary models, with a smaller number of factors, were also estimated by testing various second-degree interactions. 

The statistical software IBM SPSS Statistics for Windows, v. 21.0. (IBM Corp., Armonk, NY, USA) was used for analysis. 

## 4. Results

The average indexes of four scales relating to mental well-being are presented in Table 2. Girls obtained significantly worse results in terms of levels of depression and stress, as well as general satisfaction with life. Older students obtained better results than younger respondents in terms of coping with social situations but also reported as slightly higher in the level of stress (*p* = 0.051). Students with chronic diseases achieved worse results than their healthy peers in all four areas. The values of the two indexes measuring negative aspects of mental health significantly declined in line with the improvement of the overall assessment of the surrounding environment; with that, the levels of the indexes measuring positive aspects increased. 

Table 3 displays changes in the average mental health indexes, depending on the overall level of PA, involvement in sedentary activities, and duration of sleep. The average values of two negatively oriented indexes (CES-DC and Cohen’s stress) decreased in line with improvements in PA and length of sleep and declined with the time spent in front of a screen or with mobile devices. In the case of two positively oriented indexes, their mean values rose in the group of adolescents representing better behaviours. Only the ability to cope in social situations shows no significant correlation with the length of sleep.

Table 4 presents the estimated results of multivariate models. The general PA index shows a positive correlation with positive mental health indicators and a negative correlation with negative ones. However, in the multivariate analyses, its impact was maintained for three outcome variables, apart from the depression scale. The assessment index of the surrounding environment, duration of sleep, and gender were significant predictors in all the models. The impact of age disappeared in multivariate analyses. The impact of chronic diseases and sedentary activities was maintained in three models and disappeared for the scale related to coping in social situations. 

The 3-way interaction between gender, environment perception, and overall level of PA, as predictors of mental health, was revealed only in relation to general satisfaction with life, as measured with Cantril’s Ladder. The result described in Table 4 was confirmed by a general linear model with categorised data, where 3-way interaction was significant at *p* = 0.009. The final model explained 20.0% of variability in Cantril’s index. The average score for boys ranged from 4.59 (unsupportive environment and low PA) to 8.40 (supportive environment and high PA), while among girls the increase was from 5.22 to 7.98. The greatest differences between boys and girls were visible in the least privileged communities (Figure 1a,b). A very strong protective effect was visible only in relation to boys. Girls living in an unsupportive family, school, and/or neighbourhood environment were characterised by very low satisfaction with life indexes, despite a good level of PA. 

If we estimate similar models when analysing individual PA components, we may notice interesting interactions between behavioural and environmental factors and participation in team sports activities. This is particularly visible in models involving the determinants of the CES-DC depression index; that is, in cases where the effect of the general PA index was eliminated. One example of the accumulation of protective factors and risk factors is presented below. The 3-way interaction between gender, environment, and team sports participation appeared to be insignificant in the GLM model adjusted additionally to MVPA (*p* = 0.106). However, 2-way interaction without gender appeared to be significant (*p* = 0.009), and the effect of team sports was revealed only in this interaction. The final model explained 18.8% of CES-DC variability. 

The average depression index assumes declining values, in line with the improvement of surrounding environment, up to 7.52 in a supportive environment. The difference between those who engage in team sports and those who do not was visible in the group raised in an unsupportive neighbourhood, school, and/or family environment, the depression index being 13.49 and 15.12, respectively (Figure 2). 

It is also worth looking at the results of interactions between the features of the environment and the occurrences of chronic disease as stress predictors (Figure 3). Cohen’s stress index reached 8.99 among adolescents with chronic diseases who grew up in an unfriendly environment, as opposed to 8.51 among their healthy peers from the same environment. In the case of a supportive family, school, and neighbourhood environment, perception of stress by ill and healthy subjects was identical and much lower (5.04). The 3-way interaction between gender, environment, and chronic disease appeared to be insignificant in the GLM model (*p* = 0.461). However, 2-way interaction without gender appeared to be close to a significant level (*p* = 0.059) and the final model explained 19.2% of Cohen’s index variability. 

## 5. Discussion

The study presents information relating to a representative sample of 3693 school children aged 15 to 17 years. The focus of interest was the correlation between PA and mental health. It was verified that the protective effect of PA for mental health is maintained, when other behavioural, demographic, and environmental factors were taken into account. Data from the most recent round of the HBSC study have been used; these continue to be a valuable source of information about the mental health of schoolchildren. Poland was one of the few countries to include 17-year-olds in the study. The extended version of the questionnaire, designed specifically for older adolescents, contains various questions concerning the mental health of teenagers and the way they spend their free time. The results of HBSC studies have not confirmed that mental well-being continues to deteriorate after the 15th birthday, while the ability to cope actually improves. 

An important feature of this study is that it links the analysis of the impact of PA on adolescents’ health with both sedentary activities and duration of sleep. Kong et al. limited their study to the correlations between these behaviours, using obesity as the outcome variable. After analysing data on 53,769 teenagers, it was demonstrated that these three health determinants should be considered in parallel, because the underlying mechanisms for each of these factors are different [25]. On the other hand, within the framework of the American Youth Risk Behaviour Surveillance System (YRBSS), the percentages of teenagers cumulatively fulfilling recommendations regarding PA; admissible number of hours spent in front of the computer, mobile phone, or TV screen; and duration of sleep were assessed on the basis of a sample of 9,589 teenagers. All the recommendations were fulfilled by 11.8% boys and 5.0% girls, and none of them by 14.1% and 17.8%, respectively [26]. Our analyses indicated that all three behaviours that are discussed are independent predictors of two mental well-being indicators. While the combination of PA and sedentary activities is well described in the literature [27], duration of sleep as an additional factor only appears sporadically. The literature assumes a two-way correlation between sleep and mental health [28]. 

A large number of factors determining satisfaction with life, as measured by Cantril’s Ladder, were identified and analysed by authors in earlier research [29]. For this reason, the study may be treated as a continuation of previous research in which the connection between Cantril’s Ladder and the family and school environment was demonstrated. However, no behavioural factors were analysed in these other investigations. 

It seems that the single most important analysed indicator is depression, which more frequently translates into a clinical diagnosis. Physical activity may help individuals cope with depression, but little, as yet, is known about such mechanisms. The results of studies among persons suffering from depression indicate that taking up PA mitigates depression insofar as it affects internal motivation and the general feeling of being able to influence one’s life [30]. In all likelihood, it is not PA in itself that helps to treat depression, but the resulting and increasing feeling of self-efficacy and being a person capable of being active [31]. Cognitive Behavioural Therapy (CBT) specialists recommend individual and group PA for depression patients three times a week for 30 min. They emphasise the importance of organised classes run by a coach. They pay special attention to group classes, which strengthen internal motivation indirectly, thereby reducing depressive symptoms [32]. Among studies analysing the correlation between the PA of young people or sedentary activities and the risk of developing depression, one of particular note is the work of Sund et al. In 1998 to 2000, these authors conducted a study among 2464 teenagers (aged 12 to 15 years) in Norway. Depressive symptoms were identified on the basis of the so-called Mood and Feelings Questionnaire (MFQ); the level of PA and sedentary activities was defined by the surveyed adolescents themselves. The results obtained indicate that, for both genders, low PA is a risk factor in terms of the occurrence of depression symptoms. In boys, this effect was exacerbated by frequent sedentary activities [33]. Different results were obtained by Hume et al., who studied the influence of PA and sleep on the mental health of 15-year olds. As did our study, these authors used the CES-DC scale to determine depressive symptoms. In their analyses, however, they did not confirm a significant connection, which may be a reflection of cultural differences [34]. In our study, the association between the general PA index and the level of depression, according to CES-DC, was visible in simple average comparisons but disappeared in multivariate analysis. However, if we consider a specific aspect of PA (team sports), the interaction of PA with other depression determinants could be observed (suggesting the possible influence of environmental factors). 

Researchers agree that PA is a factor which reduces stress [35]. Unfortunately, it has been demonstrated that stress causes a reduction in the level of PA. This connection may be described by the vicious circle metaphor. Stress reduces motivation for PA and increases the level of unpleasant emotions (anxiety, sadness) which, in turn, exacerbate an already inactive lifestyle. In effect, this leads to undermining the immunological system as well as increased fatigue and physical weakness—which is tantamount to reducing the ability and willingness to take up PA [36]. The connection between stress and PA has also been analysed in the context of reactivity and resistance of the nervous system. It has been demonstrated that highly reactive individuals (i.e., those more sensitive to stimuli), with a lower ability to cope with stress, experience a higher level of anxiety in stressful situations and are more likely to experience mental health problems than healthy controls. For such persons (in comparison with less reactive individuals), PA is of particular importance in terms of reducing stress [37]. 

The inclusion of PA among stress reduction strategies requires a change of approach to PA in schools, from classes focused on winning and performance to those which include the entire group (and, thus, encourage social interaction). Altogether, the atmosphere during the sports classes is of key importance. Physical activity in everyday life should also be promoted, as opposed to someone only being active during sports classes [38]. 

Everyday PA is moderated among others by social factors. Wide-ranging analyses of the relationship between moderate PA (MVPA) among children and young people aged 10 to17 years and social factors were carried out by Heitzler et al. This group found that perceived self-efficacy and support received from parents and peers affect taking up PA by young people [39]. The effect of parental support on fulfilling recommendations for the desirable amount of exercise on weekdays and weekends was also confirmed by Czech researchers. The more active and supportive the parents, the more often the children are likely to fulfil WHO requirements concerning PA; conversely, the more time parents spend on sedentary activities, the less likely their children are to adhere to these recommendations [40]. 

The positive effect of parental and peer support on the well-being of teenagers, as well as the economic status of the neighbourhood in which they reside, was also confirmed by the studies conducted as part of the American longitudinal project ‘Child Development Supplement of the Panel Study of Income Dynamic’ [41]. 

The paper devotes much attention to environmental factors that can moderate the relationship between physical activity and mental health in adolescents. It is currently a dynamically developed approach to analyse determinants of health. This denotes shifting from demonstrating simple connections towards understanding the process and mechanism under investigation. It is difficult to find many papers that relate to moderated relationships in relation to the teen population and the impact of physical activity on mental health. In the secondary analysis of outstanding papers on this relationship, identified in seven systematic reviews, a lack of interest in environmental moderators was found [42]. However, some examples of analyses related to the adult population could be indicated [43,44]. In the longitudinal studies, some authors focused rather on the mediation effect. According to Ohrnberger et al., the relationship between previous and current self-rated health is mediated by physical activity [45]. It seems that our research fits well into the mainstream of complex (moderated) relationships, and the results obtained can have important practical implications.

Among the strengths of the current study is that it draws attention to students with chronic diseases. The presence of long-lasting health conditions significantly reduces all four mental health indicators. It was noticed, however, that the differences between healthy and ill individuals disappeared in the presence of a supportive environment. This is consistent with the previous study based on HBSC 2014 survey results, where the protective effect of social capital was demonstrated with regard to the mental health of students with chronic diseases (as reflected in analyses of KIDSCREEN and GHQ-12 indexes) [46]. It was demonstrated that teenagers with chronic diseases are more prone to depression than their healthy peers [47]. According to WHO recommendations, with regard to supporting teenagers with chronic diseases, it is important to create a friendly social environment. Depending on how it is perceived by this group of young people, the social environment may become a risk factor or a protective factor for their development. Both the physical elements of the environment (e.g., a school adapted to the special needs of a teenager with a chronic condition), as well as the social dimension (understood as the ability to build relationships and social support), are important [48]. The negative perception of the social environment of teenagers with a chronic disease increases the risk of difficulties in social functioning in adulthood, including their pursuit of further educational opportunities [49] and employment [50].

The present work has a number of limitations, among which are: difficulties managing the sheer volume of collected empirical data; relying on self-reports of young people; and the cross-sectional nature of the study. The scope of analysed factors determining mental health was wide (a total of 29 variables, partly forming indexes), but it could have been extended, for example, with such determinants as risky behaviours. Conclusions concerning well-being are based on symptoms, not on a diagnostic interview. Similarly, the measured level of PA and duration of sleep may be treated as approximations. Measurement of the still infrequent element of sleep, while offering insight, was limited to school days and, thus, was not fully representative. Furthermore, no information was available on the quality of sleep or the types of sleep disorders, if any, that may have been present. In addition, the use of combined indexes—while aimed at simplifying the analyses—may have given rise to some interpretative difficulties. Thus, in subsequent studies, more space should be devoted to a separate analysis of the impact of the school, family, and neighbourhood. While attempts in this respect have been made in our previous studies, more examination of these factors is warranted. In subsequent studies, we plan to focus more on gender differences, but at this stage of research analysis, we were limited to interactions between variables, including gender as one of the analysed indicators. Nonetheless, the large representative sample surveyed yielded important practical implications regarding the nature and presence of factors protecting young people against mental health problems. The research is based on standards of the international research network, HBSC. Thanks to these standards, we were able to analyse a number of environmental factors. This is an evident strength of the study. 

## 6. Conclusions

The mental health of young people is moderated by a series of demographic, behavioural, and environmental factors. Sedentary behaviours, insufficient amounts of sleep, and low levels of PA are clear risk factors. Activity on a physical level, including participation in organised classes, may play a protective role for the mental health of young people. Educational campaigns aimed at improving the mental health of adolescents may include aspects of promoting PA and sufficient amounts of sleep. The protective effect of the surrounding environment must also be taken into account. Subsequent studies should devote more space to a separate analysis of the influence of the school, family, and neighbourhood.

## Figures and Tables

**Figure 1 ijerph-16-00915-f001:**
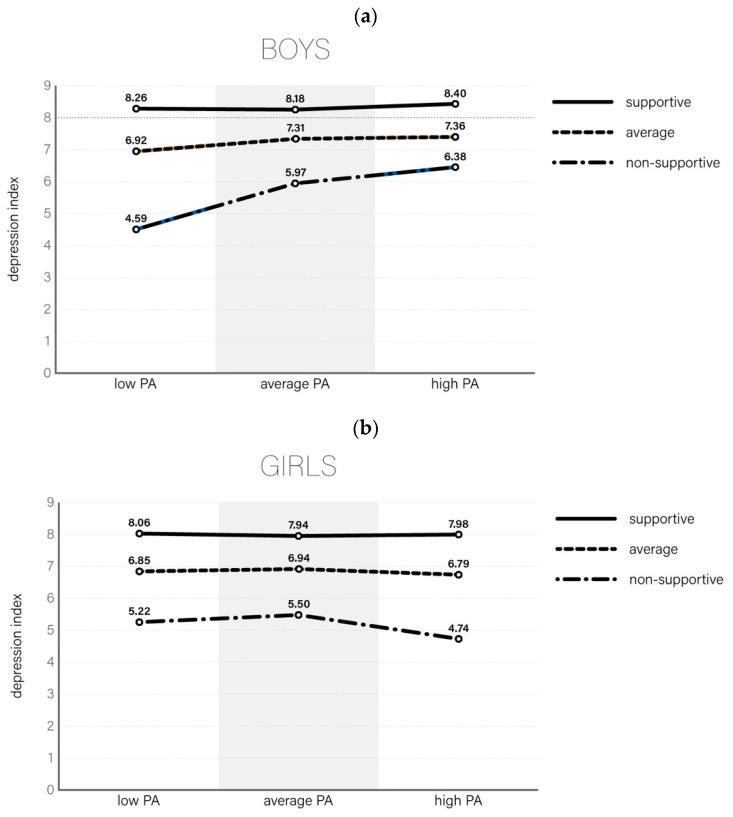
(**a**) Life satisfaction (Cantril index) among boys according to physical activity and growing-up environment; (**b**) Life satisfaction (Cantril index) among girls according to physical activity and growing-up environment.

**Figure 2 ijerph-16-00915-f002:**
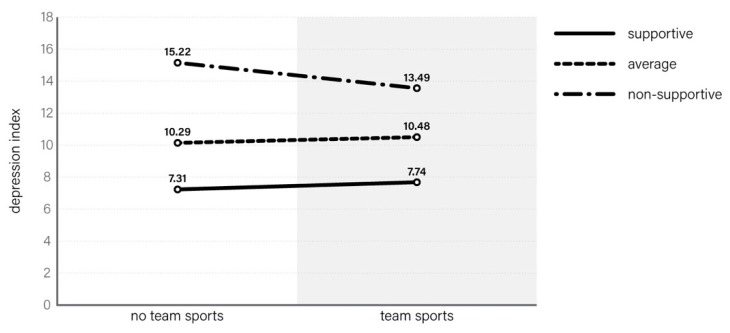
Depression level among students according to team sport participation and growing-up environment.

**Figure 3 ijerph-16-00915-f003:**
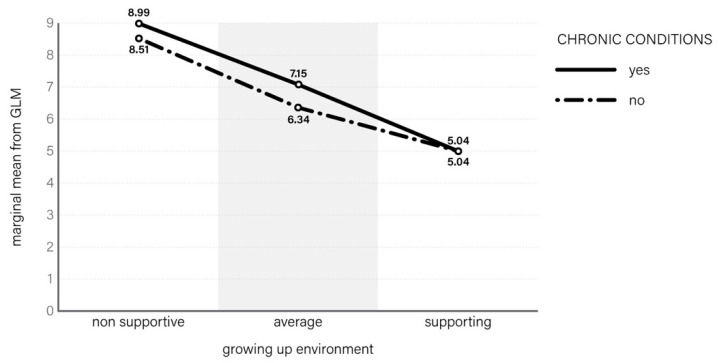
Stress level among students according to chronic-conditions status and growing-up environment.

**Table 1 ijerph-16-00915-t001:** The components of the scale describing the growing up environment.

Subscale	No. of Items	Source	Example of Item	Alfa-Cronbach *
Neighbourhood deprivation	3	HBSC protocols since 2001/02; optional in 2017/18	In the area where you live there is litter, broken glass or rubbish lying around	0.710
Neighbourhood social capital	4	HBSC protocols since 2001/02; optional in 2017/18	I could ask for help or a favour from neighbours	0.716
School climate-classmates and teacher support	6	HBSC protocols since 1990; mandatory items in 2017/18	The students in my class(es) enjoy being together; I feel a lot of trust in my teachers	0.794
Family support	4	HBSC protocols since 1990; mandatory items in 2017/18	My family really tries to help me	0.941
Family social position	1	Goodman’s adaption of McArthur scale; national item in Polish HBSC 2018 study	Imagine that this ladder pictures how Polish society is made up. Please tell us where you think your family would be on this ladder.	-

* The univariate structure of these scales was also confirmed.

**Table 2 ijerph-16-00915-t002:** Mental well-being of adolescents according to basic characteristics.

Variables	*n*	CES-DC		Cohen Stress		Cantril’s Ladder		Social Self-Efficacy	
		Mean	SD	Mean	SD	Mean	SD	Mean	SD
Total	3693	10.66	6.31	6.67	3.23	6.96	1.97	20.70	6.04
Gender									
Boys	1741	9.33	5.69	5.80	3.02	7.27	1.90	20.74	6.25
Girls	1952	11.83	6.60	7.44	3.21	6.69	1.99	20.67	5.86
*p*		<0.001		<0.001		<0.001		=0.436	
Age/grade									
15 yrs	1993	10.51	6.23	6.57	3.20	7.01	1.91	20.50	6.03
17 yrs	1700	10.83	6.40	6.78	3.26	6.90	2.03	20.93	6.05
*p*		=0.173		=0.051		=0.226		=0.017	
Health status									
Chronically ill	650	12.13	6.87	7.37	3.38	6.49	2.10	20.10	6.27
Healthy	3036	10.34	6.14	6.52	3.18	7.06	1.93	20.83	5.99
*p*		<0.001		<0.001		<0.001		=0.007	
Environment									
Unsupportive	712	15.03	6.67	8.80	3.25	5.47	2.14	17.69	6.52
Average	2140	10.31	5.75	6.52	2.96	7.07	1.75	20.74	5.64
Supportive	714	7.21	4.92	4.96	2.79	8.13	1.38	23.84	5.08
*p*		<0.001		<0.001		<0.001		<0.001	

**Table 3 ijerph-16-00915-t003:** Mental well-being of adolescents according to selected behavioural factors.

Variables	*n*	CES-DC		Cohen Stress		Cantril’s Ladder		Social Self-Efficacy	
Overall PA Index		Mean	SD	Mean	SD	Mean	SD	Mean	SD
Low	789	11.78	6.78	7.43	3.37	6.56	2.10	19.38	6.13
Average	2123	10.51	6.11	6.63	3.10	7.03	1.86	20.82	5.78
High	727	9.82	6.15	5.94	3.28	7.23	2.03	21.86	6.38
*p*		<0.001		<0.001		<0.001		<0.001	
Overall SB index									
Low	732	10.01	6.22	6.37	3.34	7.24	1.90	20.77	6.27
Average	220	10.52	6.29	6.63	3.15	6.97	1.91	20.90	5.87
High	734	11.71	6.36	7.07	3.30	6.70	2.13	20.11	6.30
*p*		<0.001		<0.001		<0.001		=0.007	
Sleep duration									
6 h or less	858	12.48	6.77	7.28	3.46	6.58	2.10	20.62	6.23
6.5–7.5 h	1805	10.50	6.13	6.61	3.12	7.07	1.88	20.93	5.91
8 h or more	983	9.33	5.83	6.20	3.10	7.14	1.91	20.42	6.07
*p*		<0.001		<0.001		<0.001		=0.077	

**Table 4 ijerph-16-00915-t004:** Determinants of mental well-being according to the generalised linear model *.

Independent Variables	CES-DC		Cohen Stress		Cantril’s Ladder		Social Self-Efficacy	
	B	*p*	B	*p*	B	*p*	B	*p*
Constant	2.781	0.000	1.968	0.000	1.980	0.000	3.066	0.000
Gender	−0.182	**0.000**	−0.189	**0.000**	0.046	**0.000**	−0.035	**0.001**
Chronic conditions	0.109	**0.000**	0.073	**0.001**	−0.053	**0.000**	−0.007	0.599
Age	0.002	0.820	0.011	0.147	−0.007	0.112	0.007	0.136
Overall index of PA	−0.013	0.219	−0.031	**0.000**	0.012	**0.014**	0.039	**0.000**
Overall index of SB	0.026	**0.012**	0.026	**0.003**	−0.015	**0.003**	−0.003	0.579
Sleep duration	−0.060	**0.000**	−0.027	**0.000**	0.010	**0.028**	−0.019	**0.000**
Overall index of environment	−0.218	**0.000**	−0.160	**0.000**	0.120	**0.000**	0.095	**0.000**
Interaction among boys	0.016	0.247	0.006	0.590	−0.021	**0.001**	0.002	0.778
Interaction among girls	−0.003	0.839	−0.014	0.231	−0.013	0.061	−0.005	0.468

* Girls and healthy students are reference category; other independent variables are continuous covariates; 3-way interaction: gender x overall index of environment x overall index of PA; B—regression parameter. Bolded: *p*-values indicate statistically significant factors.

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
