# Peer review of "Physical Activity, Sedentary Behaviours and Duration of Sleep as Factors Affecting the Well-Being of Young People against the Background of Environmental Moderators"

_ijerph, 2019, doi:10.3390/ijerph16060915_

Round 1

Reviewer 1 Report

The authors mention the influence of physical activity, sedentary behaviours and duration of sleep to the mental health of the young people. Though sample size is a little bit small, this is an important point.

However, this results may not show new and impressive evidence and because methods are not appropriate and the results may not support the author’s conclusion.

1.      The author used CES-DC, Cohen stress, Cantril ladder and Social self-efficacy for mental health of the young people. This results are collect but not impressive, because 4 scales are interrelated each other.   

2.      Physical activity, sedentary behaviours and duration of sleep are also important factors. However ,these three factors are also interrelated.

3.      The author mentions ‘Physical activity is effective even if against the background of environmental’. If the author mentions this conclusion, physical activity improves each scales, CES-DC, Cohen stress, Cantril ladder and Social self-efficacy, even under revised by environmental factor. The authors may only divide three groups, this method may seem arbitrarity.

Author Response

Reviewer's comments to author:

The authors mention the influence of physical activity, sedentary behaviours and ­­­­duration of sleep to the mental health of the young people. Though sample size is a little bit small, this is an important point. However, these results may not show new and impressive evidence and because methods are not appropriate, and the results may not support the author’s conclusion.

Author’s response:

We agree with the comment that the sample size is very important in the multivariate analysis. Our research has been carried out on 3693 adolescents, so we consider it as an appropriate and sufficient sample size due to the fact that it was homogeneous considering the age. The data come from Polish part of the well-known international study – Health Behaviour in School Aged Children and are recently collected (2018). In the description of the limitations of the study we have underlined its cross-sectional character. The longitudinal or intervention study could give better evidence, but the latter are usually carried out on smaller samples and it’s difficult to take into account environmental factors in such a type of the research. However, we have now extended the explanation about strengths and weaknesses of our study.

Reviewer's comments to author:

The author used CES-DC, Cohen stress, Cantril ladder and Social self-efficacy for mental health of the young people. These results are collect but not impressive, because 4 scales have interrelated each other.   

Author’s response:

Our intention was to present four models without showing mutual interactions between them. Such approach allowed presenting both positive and negative indicators and different aspects of adolescents’ mental health.

Reviewer's comments to author:

Physical activity, sedentary behaviours and duration of sleep are also important factors. However, these three factors are also interrelated.

Author’s response:

We absolutely aware of this fact and that is why multivariate logistic regression models were estimated to identify most important predictors of four dependent variables (Tab. 4).

Reviewer's comments to author:

The author mentions ‘Physical activity is effective even if against the background of environmental’. If the author mentions this conclusion, physical activity improves each scale, CES-DC, Cohen stress, Cantril ladder and Social self-efficacy, even under revised by environmental factor. The authors may only divide three groups, this method may seem arbitrarily.

Author’s response:

According to the results presented in table 4 indexes concerning environmental indicators were treated as continuous scales (s. c. covariates). The division into three groups (non-supportive, average and supportive) was used only for the purpose of graphic presentation. The type of variables is underlined in the table’s footnote. The division into those three groups is based on standardized indexes and the distribution of quintiles (Q1; Q2-4; Q5) which is often used approach.

Reviewer 2 Report

Re: ijerph-448063

This study aims to evaluate behavioural and environmental factors on mental well-being of young people. It was found that a range of these factors were significant predictors of mental health indicators. My comments are as followed.

Material and Methods

1.     The data of the study seemed to be nested data, i.e. participants were organized at more than one level (students were sampled from classes and classes were sampled from schools). The authors may want to consider controlling for the possible clustering effects from the class/school level.

2.     In line 163, the authors claimed that they used factor analysis (Principal Components Analysis; PCA). However, factor analysis (FA) and PCA are two different analyses although they are very similar in many ways. The authors may want to clarify which analysis they actually used.

3.     The authors claimed that they used FA/PCA to develop indexes for PA, sedentary activities and the surrounding environment. However, it seemed that the sedentary activities index was calculated based on time. Did they need FA/PCA to create this index?

Results

1. Line 221, typo: The average values “of for” analysed indexes…

2. In line 222, it is confusing to describe that all the indexes rose or decline because two of the indexes represented the negative aspects of mental health and the other two represented the positive aspects. The directions of increase or decrease of the scores for these two groups of indexes are opposite.

3. The authors said, in the Methods of analysis, that the third-degree interaction: gender*general surrounding environment index*general PA index is the most important topic of the study. However, it is not easy to understand and interpret the results of a 3-way interaction because there would be 18 (2*3*3 in their case) kinds of conditions. The authors may want to make it clearer when presenting the results of 3-way interaction on page 8. From the bottom two rows of table 4 on page 8, it seemed that they examined the environment*PA interaction separately for boys and girls. Then they showed the condition in the unsupportive environment in Fig.1. But there are still other conditions in the average and supportive environments that they did not mention.

4. In line 246, (“supportive activity” and high PA) should be (“unsupportive environment” and high PA) if I understand it right.

5. The authors presented three other interactions from page 8 to 9. For the first interaction on page 8, there is no table or figure to show the result. It is not easy to follow only from text. For the latter two interactions on page 9, no statistical values were reported for these interaction analyses. In line 267, the reported value of 7.54 was not found in Fig.2.

Discussion

1. There is a lack of a summary of the key findings from the study. And the linkage of the discussion between the findings from the study should be strengthened.

2. The discussion mostly focused on the effects of PA on mental health. However, a major part of the results showed the interaction effects. The discussion about interaction is little.

3. It is surprising that the authors mentioned chronic diseases was the strength of the study. It does not seem the focus of the study and not be introduced nor discussed in previous sections in the manuscript.

Author Response

Warsaw, 9th February 2019

Dear Reviewer,

We are very grateful for the constructive comments on our paper “Physical activity, sedentary behaviours and duration of sleep as factors affecting the mental health of young people against the background of environmental moderators”

­

Based on your valuable comments, we made changes in the manuscript that are marked with yellow in the submitted revised paper. We hope that these improvements will make the paper acceptable for publication in “International Journal of Environmental Research and Public Health,”. Point-to-point answers to Your comments are provided below:

  Reviewer's comments to author:

The data of the study seemed to be nested data, i.e. participants were organized at more than one level (students were sampled from classes and classes were sampled from schools). The authors may want to consider controlling for the possible clustering effects from the class/school level.

Author’s response:

Indeed, in many studies based on HBSC data multilevel modeling is used due to its sampling design and hierarchical data structure. In this research our decision on this matter was based on the ICC indicator for dependent variables. ICC varied from 2.03% considering adolescents’ self-esteem to 3,21% for life satisfaction (Cantrill Ladder scale). In “Material and methods’ section (verse 115-116), we have given the number of schools and classes included in the study.  In our opinion, including multilevel models it is not necessary due to a large and varied sample and low ICC indicators.

           Reviewer's comments to author:               

In line 163, the authors claimed that they used factor analysis (Principal Components Analysis; PCA). However, factor analysis (FA) and PCA are two different analyses although they are very similar in many ways. The authors may want to clarify which analysis they actually used.

The authors claimed that they used FA/PCA to develop indexes for PA, sedentary activities and the surrounding environment. However, it seemed that the sedentary activities index was calculated based on time. Did they need FA/PCA to create this index?

Author’s response:

In the paper we have used PCA - Principal Components Analysis. We have now deleted the phrase: factor analysis (verse 163). FA has not been used in sedentary behaviors index neither.

Firstly, we have calculated time spent on SB (weighted average of school days and weekends, as its described in the paper verse 177) and then we have developed one aggregate standardized index for all three types of behaviours. It has the mean=0 and SD=1.

Reviewer's comments to author: Line 221, typo: The average values “of for” analysed indexes…

In line 222, it is confusing to describe that all the indexes rose or decline because two of the indexes represented the negative aspects of mental health and the other two represented the positive aspects. The directions of increase or decrease of the scores for these two groups of indexes are opposite.

Author’s response: Thank you very much it was typo in line 221. We have also changed the text describing positive and negative aspects of mental health (verse 221).

Reviewer's      comments to author:

The authors said, in the Methods of analysis, that the third-degree interaction: gender*general surrounding environment index*general PA index is the most important topic of the study. However, it is not easy to understand and interpret the results of a 3-way interaction because there would be 18 (2*3*3 in their case) kinds of conditions. The authors may want to make it clearer when presenting the results of 3-way interaction on page 8. From the bottom two rows of table 4 on page 8, it seemed that they examined the environment*PA interaction separately for boys and girls. Then they showed the condition in the unsupportive environment in Fig.1. But there are still other conditions in the average and supportive environments that they did not mention.

Author’s response:

Presented indexes in the model in Table 4 are continuous (included as covariates). They are not categorized so there are less options for factor combinations. We have now changed the Table’s 4 layout. In the last two rows, we have presented interactions between physical activity and environment separately for both genders. We have clarified the description.   

At first, in Fig. 1 we have showed only non-supportive environment. It actually could be misleading when taking into consideration other values presented in the paper and assumption about 3-way interaction. We have now changed this figure and split it to part a-boys and part b-girls.  

Reviewer's      comments to author:

In line 246 (“supportive activity” and high PA) should be (“unsupportive environment” and high PA) if I understand it right.

Author’s response: We have changed this text accordingly.

Reviewer's      comments to author:

The authors presented three other interactions from page 8 to 9. For the first interaction on page 8, there is no table or figure to show the result. It is not easy to follow only from text. For the latter two interactions on page 9, no statistical values were reported for these interaction analyses. In line 267, the reported value of 7.54 was not found in Fig.2.

Author’s response: This part of the text has now been significantly changed. The number of presented interactions has been reduced and all are illustrated in numbers, with p-value given. In each of them the correlation with gender has been checked.

Kind regards

Dorota Kleszczewska and co-authors

Reviewer 3 Report

The paper describes a cross sectional study of secondary school students in Poland who were surveyed using a screening instrument for depressive symptoms, a measure of stress, a measure of life satisfaction and a measure of self-efficacy and their relationship to physical activity, sedentary behaviours, length of sleep and perception of the environment.  Covariates in the analyses were gender, age, and chronic disease.  Most findings were unsurprising. Of significance, physical activity was most protective for those who lived in less supportive environments.   The finding lends strong support in my opinion for engaging young people from disadvantaged backgrounds in organised sport and other physical activity such as dance.  The paper appears in the context of the crowded space as there is an extensive literature on the relationship between lifestyle factors and mental wellbeing including larger studies and more robust longitudinal studies.   The authors were ambitious in the scope of their study and attempted to manage a complex data set.  However, there were important omissions from the life style factors such as diet and substance use.   One might also wonder why the mental health and wellbeing outcomes were limited to depression stress general satisfaction with life and self-efficacy.   That said, the paper makes a valuable contribution by validating previous research in a new population and by examining the interaction between lifestyle behaviours and environment.

I have only a few comments:

I think the term ‘wellbeing’ would be preferable to ‘mental health’ in the title, as it more accurately characterizes the content of the paper.

There were marked gender differences. Given the known gender differences in rates of depression in adolescence, and the plausible possibility that lifestyle factors for males and females interact somewhat differently with wellbeing I think it would be preferable to report all analyses separately for males and females. By using gender as a covariate in the analyses the authors could well be missing some important associations.  

Author Response

Warsaw, 9th February 2019

Dear Reviewer,

We are very grateful for the constructive comments on our paper “Physical activity, sedentary behaviours and duration of sleep as factors affecting the well-being of young people against the background of environmental moderators”

­

Based on your valuable comments, we made changes in the manuscript that are marked with yellow in the submitted revised paper. We hope that these improvements will make the paper acceptable for publication in “International Journal of Environmental Research and Public Health,”. Point-to-point answers to Your comments are provided below:

We are very grateful for such a warm reception of our paper.  We do appreciate the Reviewer’s interest in the field of our research – the physical activity and environmental indicators’ correlations. We are aware that our results are in line with general expectations, but the aim of the study was to gain particular numerous results specifically for the adolescent population. Planning the research, we were not exactly sure if determinants such as sleep duration or sedentary behaviours would be accepted in final models and if interference will occur consistently.

Reviewer's      comments to author:

I think the term ‘wellbeing’ would be preferable to ‘mental health’ in the title, as it more accurately characterizes the content of the paper.

Author’s response: The title has been changed accordingly

Reviewer's      comments to author:

There were marked gender differences. Given the known gender differences in rates of depression in adolescence, and the plausible possibility that lifestyle factors for males and females interact somewhat differently with wellbeing I think it would be preferable to report all analyses separately for males and females. By using gender as a covariate in the analyses the authors could well be missing some important associations.  

Author’s response:

The comment on gender differences is particularly important but proceeding separate analysis for boys and girls would significantly increase the volume of the paper. In the revised version of the paper we used 3-way interaction where gender is one of indicators. In case there was lack of gender determinant the analysis was limited to 2-way interaction.

The following corrections have been introduced:

-          In the part about limitations of the study either gender differences or the lack of other life style determinants have been underlined. (verse 421)

-          The figures (Fig 1, verse 256) and their descriptions have been changed accordingly

Kind regards

Dorota Kleszczewska

This manuscript is a resubmission of an earlier submission. The following is a list of the peer review reports and author responses from that submission.